# TORC2-Dependent Ypk1-Mediated Phosphorylation of Lam2/Ltc4 Disrupts Its Association with the β-Propeller Protein Laf1 at Endoplasmic Reticulum-Plasma Membrane Contact Sites in the Yeast *Saccharomyces cerevisiae*

**DOI:** 10.3390/biom10121598

**Published:** 2020-11-25

**Authors:** Magdalena Topolska, Françoise M. Roelants, Edward P. Si, Jeremy Thorner

**Affiliations:** 1Division of Biochemistry, Biophysics and Structural Biology, Department of Molecular and Cell Biology, University of California, Berkeley, CA 94720-3202, USA; magdatopolska94@gmail.com (M.T.); roelants@berkeley.edu (F.M.R.); edwardsi5197@gmail.com (E.P.S.); 2Villum Center for Bioanalytical Sciences, Department of Biochemistry and Molecular Biology, University of Southern Denmark, 5000 Odense, Denmark; 3Eastern Virginia Medical School, P.O. Box 1980, Norfolk, VA 23501-1980, USA

**Keywords:** membrane contact sites, ergosterol, regulation, protein kinases, homeostasis

## Abstract

Membrane-tethered sterol-binding Lam/Ltc proteins localize at junctions between the endoplasmic reticulum (ER) membrane and other organelles. Two of the six family members—Lam2/Ltc4 (initially Ysp2) and paralog Lam4/Ltc3—localize to ER-plasma membrane (PM) contact sites (CSs) and mediate retrograde ergosterol transport from the PM to the ER. Our prior work demonstrated that Lam2 and Lam4 are substrates of TORC2-regulated protein kinase Ypk1, that Ypk1-mediated phosphorylation inhibits their function in retrograde sterol transport, and that PM sterol retention bolsters cell survival under stressful conditions. At ER-PM CSs, Lam2 and Lam4 associate with Laf1/Ymr102c and Dgr2/Ykl121w (paralogous WD40 repeat-containing proteins) that reportedly bind sterol. Using fluorescent tags, we found that Lam2 and Lam4 remain at ER-PM CSs when Laf1 and Dgr2 are absent, whereas neither Laf1 nor Dgr2 remain at ER-PM CSs when Lam2 and Lam4 are absent. Loss of Laf1 (but not Dgr2) impedes retrograde ergosterol transport, and a *laf1∆* mutation does not exacerbate the transport defect of *lam2∆ lam4∆* cells, indicating a shared function. Lam2 and Lam4 bind Laf1 and Dgr2 in vitro in a pull-down assay, and the PH domain in Lam2 hinders its interaction with Laf1. Lam2 phosphorylated by Ypk1, and Lam2 with phosphomimetic (Glu) replacements at its Ypk1 sites, exhibited a marked reduction in Laf1 binding. Thus, phosphorylation prevents Lam2 interaction with Laf1 at ER-PM CSs, providing a mechanism by which Ypk1 action inhibits retrograde sterol transport.

## 1. Introduction

In eukaryotic cells, sterols are essential constituents of the plasma membrane (PM) [1], contributing to PM fluidity, rigidity, thickness, and permeability [2], partitioning into discrete domains [3,4], and supporting the activities of membrane proteins vital for many cellular processes [3,5]. The PM of budding yeast (*Saccharomyces cerevisiae*) is highly enriched in ergosterol, the most abundant sterol in this species [6,7]. After its de novo synthesis in the endoplasmic reticulum (ER) [8,9], ergosterol is delivered to the PM and other target organelles largely by non-vesicular transport executed by lipid transfer proteins (LTPs) [10,11,12]. Current evidence indicates that this inter-organellar lipid exchange occurs at locations where the two membranes involved are closely apposed, referred to as membrane contact sites [13,14], or here, for short, contact sites (CSs).

One family of LTPs located at CSs are members of a class of sterol-binding proteins tethered to a membrane by a single C-terminal transmembrane segment, which have been designated both LAM (Lipid transfer protein Anchored at a Membrane contact site) or LTC (Lipid Transfer at a Contact site) [15,16,17]. The six members of the *S. cerevisiae* LAM/LTC family fall into three pairs of paralogs: Lam1/Ysp1 and Lam3/Sip3, Lam2/Ltc4/Ysp2 and Lam4/Ltc3, and Lam5/Ltc2 and Lam6/Ltc1. All LAM/LTC proteins bind ergosterol within a StARkin (or StART-like) domain, so named because of the kinship of its structure to that of the sterol-binding element in the mammalian steroidogenic acute regulatory transfer protein family [18,19,20]. N-terminal to their StARkin domain, all family members also possess a pleckstrin homology (PH)-like domain [21]. Lam2 (1438 residues) and Lam4 (1345 residues) are unique among the yeast LAM/LTC family in being the largest, in possessing two tandem StARkin domains [21,22,23], and in being localized exclusively at ER-PM CSs, i.e., not at junctions between the ER and any other membrane [15,16,17]. In addition to sterol trafficking, ER-PM CSs also have been implicated, to date, as sites for lipid synthesis [24], Ca^2+^ flux [25], and autophagosome initiation [26].

The StARkin domains of Lam2 and Lam4 in both their apo form and sterol-bound state have been determined at atomic-scale resolution using both X-ray crystallography and NMR [21,22,23,27]. Each StARkin domain has a deep cup-like or half-barrel-shaped structure with a body composed of three α-helices, the longest of which (α3) is enwrapped by a six-stranded antiparallel β-sheet (helix- β-grip fold), that is capped by a lid-like flexible loop (Ω_1_ between strands β_2_ and β_3_). This structure creates an internal hydrophobic cavity in which ergosterol binds in a “head down” orientation, i.e., with its C3 hydroxyl group contacting the bottom of the barrel (where the -OH interacts with the protein via a network of H_2_O molecules) and its C17-linked branched C_9_ tail positioned at the entrance to the cavity (with the lid closed over it). Thus, an open conformation of the Ω1-loop is essential for ergosterol entry and egress. At least in the second StARkin domain of Lam4, there is a patch of basic residues on the surface of the lid, suggesting a model [23] wherein lid opening could occur when it engages the anionic surface of a membrane, allowing the sterol to slip into the hydrophobic cavity “head-first”, followed by closure of the lid upon dissociation of the protein from the membrane (and reversal of these steps would permit sterol exit) [28].

Also, immediately C-terminal to the second StARkin domain in both Lam2 and Lam4 (and just ~40 residues upstream from their C-terminal transmembrane ER anchor), there is a sequence segment that is highly positively charged (10 basic residues out of 20) [17]. It has been proposed [21] that this polybasic tract might help promote close apposition between these ER-tethered proteins and the cytosolic leaflet of the PM, which is enriched in acidic glycerophospholipids, such as PtdSer and PtdIns4,5P_2_ [29]. The PH domains in the LAMs/LTCs are of the so-called GRAM (glucosyltransferases, Rab-GTPase activating proteins, and myotubularins) sub-type, which share only modest sequence homology with canonical PH domains [30,31]; yet, the overall topology of the PH domain from Lam6/Ltc1 [16], which is the only one yet studied in detail at the structural level [21], exhibits a typical PH domain topology with an additional, short N-terminal amphipathic α-helix. Unlike a typical PH domain that contains positive charges at the ends of its β-sheets, which serve as its contacts to the head group of a phosphoinositide [32], the Lam6 PH domain has, instead, many hydrophobic residues exposed at the ends of its β-sheets, forming hydrophobic patches on the surface [21]. Moreover, when modeled on the Lam6 PH domain, the loop residues at the ends of the β-sheets in the PH domains of the other LAM/LTC members are quite variable, suggesting specificity in the membrane and/or protein partners with which they interact.

Despite these insights about their structure, little was known about how the function of Lam2 and Lam4 in sterol transport is regulated. However, in yeast, maintenance of homeostasis for all other well-established PM constituents (glycerophospholipids, sphingolipids, and integral membrane proteins) is under the control of the TORC2-activated protein kinase Ypk1 (and its paralog Ypk2) [33,34,35]. Ypk1 executes its regulatory function through direct phosphorylation of enzymes and other effectors involved in controlling PM lipid and protein composition. Hence, it was perhaps not surprising that in our comprehensive search for Ypk1 substrates [36], Lam2 (and, hence, also its paralog Lam4) was identified as a high-confidence target of Ypk1. Our subsequent work confirmed that both Lam2 and Lam4 are indeed direct and physiologically relevant substrates of Ypk1 [37]. We demonstrated that Ypk1-mediated phosphorylation inhibits the ability of Lam2 and Lam4 to promote retrograde transport of sterol from the PM to the ER, thereby retaining ergosterol in the PM under conditions that stimulate TORC2-mediated activation of Ypk1, which we found aids cell survival under such stresses [37].

Here, we explore the molecular mechanism by which Ypk1-mediated phosphorylation impedes sterol transport by Lam2 and Lam4 at ER-PM CSs. We investigated, first, whether Ypk1-specific phosphorylation affects the localization or stability of Lam2 and Lam4. Then, based on a previous report that identified a predicted, but poorly characterized, WD40 repeat-containing protein (Ymr102c) to co-purify with immuno-enriched green fluorescent protein (GFP)-tagged Lam2 [38], we examined the function of this protein and its paralog (Dgr2) and their association with Lam2 (and Lam4) both at ER-PM CSs in vivo and as recombinant proteins in vitro. WD40 repeat proteins adopt a β-propeller fold with an overall toroid-shaped topology [39], and these domains often are involved in mediating protein–protein interactions within both the yeast and the human interactomes [40,41,42,43,44]. Based on our findings, the designated gene name in the *Saccharomyces* Gene Database for *YMR102c* is now *LAF1* (for Lam2/Ltc4-Associated Factor). Finally, we examined how phosphorylation of Lam2 by Ypk1 influences its ability to interact with Laf1.

## 2. Materials and Methods

### 2.1. Construction of Yeast Strains and Growth Conditions

All *S. cerevisiae* strains in this study (Table 1) were constructed using standard methods for genetic manipulation of yeast [45]. Yeast strains were grown routinely at 30 °C on standard rich (YP) or defined minimal media containing 2% glucose as carbon source (SCD) and supplemented with appropriate nutrients to maintain selection for plasmids.

### 2.2. Plasmids and Recombinant DNA Methods

All plasmids in this study (Table 2) were constructed in *Escherichia coli* DH5α strain using standard recombinant DNA procedures [49]. All constructs were verified by nucleotide sequence analysis at the UC Berkeley DNA Sequencing Facility. All PCR reactions were performed using Phusion^TM^ High-Fidelity DNA Polymerase (New England Biolabs, Inc., Ipswich, MA, USA) and site-directed mutagenesis was performed using appropriate mismatch oligonucleotide primers using the QuickChange^TM^ protocol (Agilent Technologies, Inc., Santa Clara, CA, USA), all according to the recommendations of these suppliers.

### 2.3. Production and Purification of GST-Fusion Proteins

*E. coli* BL21(DE3) cells were transformed with appropriate plasmids expressing the desired protein or protein fragment and grown at 37 °C overnight in LB media under appropriate conditions to maintain selection for the drug-resistance marker on the plasmid. The resulting cultures were diluted to A_600 nm_ = 0.2 and grown at 30 °C up to A_600 nm_ = 0.6, whereupon isopropyl β-D-1-thiogalactopyranoside (IPTG) was added (final concentration 0.6 mM) and the cells were incubated either for an additional 3 h at 30 °C (for cells containing the plasmid expressing Lam2(482–1259)) or overnight at 16 °C (for cells containing the plasmid expressing Laf1). Protein expression was verified by SDS-PAGE and staining with Coomassie blue dye. The resulting cultures were harvested by centrifugation, washed once with phosphate-buffered saline (PBS), and recollected by centrifugation. The resulting pellets were snap-frozen in liquid N_2_ and stored at −80 °C prior to initiating protein purification.

For protein purification, the cell pellets were thawed on ice and resuspended in 10 mL ice-cold lysis buffer (0.5% Tween-20, 1 mM EDTA, 2 mM MgCl_2_ in PBS) for every 50 mL of original culture. Lysozyme (final concentration 0.2 mg/mL) was added, and the suspension was incubated for 20 min on ice. Cell rupture was then completed by two 15 s bursts of sonication on ice, with a 30 s rest period on ice between the two bursts. To the resulting lysate, dithiothreitol (DTT; final concentration 1 mM) was added, and then the extract was clarified by centrifugation at 1.2 × 10^4^× *g* for 10 min at 4 °C. The resulting supernatant solution was transferred to a fresh Falcon tube and 100 µL of glutathione S-transferase (GST) Sepharose™ 4B beads, pre-washed in PBS, were added. After incubation for 1 h with gentle agitation on a rollerdrum in a 4 °C room, the GST-agarose beads were collected by centrifugation and washed three times by successive resuspension and recentrifugation in wash buffer (0.1% Tween-20, 0.1 mM DTT in PBS). To elute the bead-bound GST-fusion protein and simultaneously remove its GST tag, the following procedures were used. For GST-fusion proteins expressed from pGEX6P-1, the protein-bound resin was washed once with PreScission^TM^ Protease cleavage buffer (200 mM NaCl, 1.5 mM Mg-acetate, 10 mM NaF, 10 mM β-glycerol-phosphate, 1 mM DTT, 50 mM Tris-HCl (pH 7.5)) and incubated overnight at 4 °C in 40 µL of the same buffer containing 3 µL (2 units/μL) PreScission^TM^ Protease (GE Life Sciences, Chicago, IL, USA) (for every 100 µL of resin). For GST-fusion proteins expressed from pKM263, the protein-bound resin was washed once with tobacco etch virus (TEV) protease cleavage buffer (0.5 mM EDTA, 1 mM DTT, 50 mM Tris-HCl (pH 8)) and incubated overnight at 4 °C in 80 µL of the same buffer containing 1 µL (10 units/μL) GST-TEV Protease (Sigma Aldrich, St. Louis, MO, USA) (for every 100 µL of resin). After the overnight incubation, the beads were removed by centrifugation and the resulting supernatant solution containing the affinity-purified protein was transferred to a fresh Eppendorf tube, which was kept on ice prior to use. Recovery of the protein of interest was verified by SDS-PAGE and staining with Coomassie blue dye and the purified proteins were always prepared fresh prior to each experiment.

### 2.4. Production and Purification of Ypk1 from S. cerevisiae

To purify protein kinase Ypk1 [51], *S. cerevisiae* strain CGA84 (*ura3*) expressing the Gal4-human estrogen receptor herpes simplex virus transactivator fusion protein (GEV) [46] was transformed with *URA3*-marked plasmid BG1805 expressing Ypk1 with a C-terminal His_6_-HA-3C-ZZ tag from the *GAL1* promoter and grown overnight in SC-Ura medium. Samples of the overnight starter culture were diluted to A_600 nm_ = 0.02 in four separate 1 L volumes of SC-Ura, which were grown at 30 °C to A_600 nm_ = 0.4, and then induced by addition of 20 µM β-estradiol (20 µM final concentration) and incubation continued overnight. The resulting cultures were chilled on ice and the cells harvested by centrifugation at 4 °C. The resulting cell pellets were washed by resuspension in TAB-B* buffer (200 mM NaCl, 1.5 mM Mg-acetate, 1 mM DTT, 2x protease inhibitor full tablets (Roche Diagnostics GmbH, Manheim, Germany), 2 mM NaVO_4_, 10 mM NaF, 10 mM Na-PPi, 10 mM β-glycerol-phosphate, 50 mM Tris-HCl (pH 7.5)) (2 mL for every 1 g of cell pellet), frozen dropwise in liquid N_2_, and the resulting frozen globules were stored at −80 °C prior to cell lysis. Cells were ruptured cryogenically in a Mixer Mill (Model MM301, Retsch GmbH, Haan, Germany), as follows. The frozen globules were placed in the mill under liquid N_2_ and pre-cooled for 5 min, and then subjected to ten 2-min runs of milling (at 15 cycles per second), interspersed with 1-min periods of cooling between each run (to evenly disperse the resulting frozen cell powder to ensure uniform cell breakage, the canister holding the frozen lysate was inverted every three cycles). The final frozen powder of cell lysate was transferred to a Falcon tube and stored at −80 °C prior to use.

For purification, samples of the frozen lysate were suspended in ice-cold TAB-B* (2 vol per g cell powder), thawed at room temperature, then placed on ice, and the major debris removed by centrifugation at 4 °C for 20 min at 1.5 ×10^4^× *g* in the SS-34 rotor of a Sorvall preparative centrifuge. The resulting supernatant solution was transferred into 16 mm × 76 mm polycarbonate ultracentrifuge tubes and clarified by centrifugation at 4 °C for 1 h in a 50Ti rotor at 3.35 × 10^4^ rpm in a Beckman L80-M ultracentrifuge. The resulting clarified extract beneath the floating lipid cake was withdrawn with a glass Pasteur pipette and transferred to a fresh Falcon tube on ice. To this extract, non-ionic detergent NP40 (0.15% final concentration) was added and then 1.25 mL of a 50:50 slurry of Sepharose-IGG beads (GE Healthcare, Chicago, IL, USA) that had been pre-washed into TAB-B2 (200 mM NaCl, 1.5 mM Mg-acetate, 0.15% NP40, 50 mM Tris-HCl (pH 7.5)) was added. After incubation for 1 h with gentle agitation on a rollerdrum in a 4 °C room, the entire solution was transferred with a pipette into a small glass column (20 mm diameter), yielding a resin bed of ~0.6 mL. The flow-through was collected and passed back over the column again to maximize the capture of the Ypk1-(His)_6_-HA-3C-ZZ fusion protein by the IgG-beads. Contaminants were then removed from the beads by four 5 mL washes of the column with Post-IGG buffer (200 mM NaCl, 1.5 mM Mg-acetate, 1 mM DTT, 0.01% NP40, 2 mM NaVO_4_, 10 mM NaF, 10 mM Na-PPi, 10 mM β-glycerol phosphate, 50 mM Tris-HCl (7.5)). After draining, the moist beads were equilibrated with 5 mL of cleavage buffer (200 mM NaCl, 1.5 mM Mg-acetate, 1 mM DTT, 0.01% NP40, 10% glycerol, 2 mM NaVO_4_, 10 mM NaF, 10 mM Na-PPi, 10 mM β-glycerol-phosphate, 50 mM Tris-HCl (pH 7.5)), and drained again. In a separate Eppendorf tube, 30 µL of PreScission Protease (2 units/μL; GE Life Sciences) was diluted into 800 µL of the same cleavage buffer. To remove the ZZ portion of the -(His)_6_-HA-3C-ZZ tag and thereby elute the purified Ypk1 from the beads, the entire diluted protease solution was transferred to the moist beads in the column, which was then tightly sealed at both ends with Parafilm™ and incubated overnight with gentle agitation on a rollerdrum in a 4 °C room. After the cleavage reaction, the liquid in the column was drained into a siliconized Eppendorf tube. To separate the purified Ypk1 from the protease (which is GST-tagged), a one-tenth volume of a 50:50 slurry of glutathione-agarose beads that had been pre-equilibrated in the same cleavage buffer was added. After incubation for 2 h with gentle agitation on a rollerdrum in a 4 °C room, the beads were removed by centrifugation for 10 min at maximum rpm in a microfuge, and the resulting Ypk1-containing supernatant solution was removed with a micropipette, and dispensed, as separate aliquots (~20 mL), into fresh siliconized Eppendorf tubes, which were snap-frozen in liquid N_2_ and stored at −80 °C prior to use.

### 2.5. Preparation of Ypk1-Phosphorylated Lam2 Fragment

Either intact GST-Lam2(482–1259)-(His)_6_ bound to beads (and pre-equilibrated in 1X kinase buffer) or purified soluble Lam2(482–1259)-(His)_6_ that had been cleaved off the beads were incubated with purified Ypk1-(His)_6_-HA, in the absence and presence of ATP. In such reactions, the amount of substrate present was in 4-fold excess over the amount of enzyme present. When bead-bound GST-Lam2(482–1259)-(His)_6_ was the substrate, the final reaction volume was 400 µL and appropriate volumes of a 5X concentrated stock of kinase buffer were added (625 mM K-acetate, 60 mM MgCl_2_, 2.5 mM EDTA, 10 mM DTT, 5% glycerol, 1 mg/mL BSA, 62.5 mM β-glycerol-phosphate, 5 mM NaVO_4_, 1x protease inhibitor cocktail mini-tablet (Roche), 250 mM HEPES (7.4)) and of a 5X concentrated stock of freshly prepared ATP solution (10 mM (pH 8)), and then H_2_O was added, and the mixtures were incubated at 30 °C for 4–7 h with gentle agitation on a rollerdrum. When purified soluble GST-Lam2(482–1259)-(His)_6_ was the substrate, the final reaction volume was 160 µ and the mixtures were incubated at 30 °C for 17 h. After phosphorylation of bead-bound GST-Lam2(482–1259)-(His)_6_, the beads were washed 3 times with wash buffer (0.1% Tween-20, 0.1 mM DTT in PBS) to remove the kinase buffer components followed by a single wash with PreScission protease cleavage buffer (200 mM NaCl, 1.5 mM Mg-acetate, 10 mM NaF, 10 mM β-glycerol-phosphate, 1 mM DTT, 50 mM Tris-HCl (pH 7.5)). The washed beads were resuspended in 200 µL of the same buffer and then 12 µL PreScission protease (2 units/µL; GE Life Sciences) was added, followed by incubation overnight at 4 °C with gentle agitation on a rollerdrum.

To assess whether exhaustive incubation with Ypk1 converted all of the resulting Lam2(482–1259)-(His)_6_ to the phosphorylated state, samples of the final reaction products were resuspended in 5X SDS-PAGE sample buffer containing 8 M urea, which were heated for 5 min at 42 °C in a water bath, resolved by electrophoresis for 1.5 h at 160 V on an 8% acrylamide containing a final concentration of 35 µM PhosTag^TM^ reagent [52], electroblotted onto a nitrocellulose membrane, and visualized by immunostaining using anti-(His)_6_ antibodies. The radiochemical assay used to measure the Ypk1-catalyzed incorporation of phosphate from [γ-^32^P]ATP into substrate proteins has been described in detail elsewhere [37,50].

### 2.6. In Vitro Binding Assay

Purified recombinant GST-Laf1 bound to glutathione-agarose beads was equilibrated with binding buffer (81 mM NaCl, 4.5 mM KCl, 1.35 mM MgCl_2_, 0.09% Triton X-100, 0.1 mg/mL BSA, 20 mM HEPES (pH 7.5)) by four successive resuspensions and recentrifugations. An equal volume (20 µL) of the final slurry of beads was distributed into each assay tube containing 300 µL of binding buffer and then a solution (20 µL) of the protein to be tested for interaction was added. After incubation for 1 h with gentle agitation on a rollerdrum in a 4 °C room, the beads were washed four times with 0.1% Tween-20, 5 mM KCl, 1.5 mM MgCl_2,_ 20 mM HEPES (pH 7.5), and, to the final wet pellet, 4 µL of 5X SDS-PAGE sample buffer containing 8 M urea was added. Then, to ensure complete elution of all proteins from the beads, they were heated for 5 min at 42 °C and samples of the resulting supernatant solutions (and, for comparison, samples representing 10% of the amount of the input of the same proteins) were then resolved by electrophoresis on either 8% or 12% acrylamide gels, transferred to a nitrocellulose membrane, and visualized by immunoblotting with appropriate antibodies.

### 2.7. Immunoblotting

Samples resolved by either SDS-PAGE (8% or 12% acrylamide) for the binding assays, or in the presence of PhosTag^TM^ reagent (Wako Pure Chemical Industries, Ltd., Osaka, Japan) for the phosphorylation reactions, were transferred onto nitrocellulose membranes and blocked in Odyssey^TM^ blocking buffer (OB) (Li-Cor Biosciences, Lincoln, NE, USA) for 1 h at room temperature, prior to immunoblotting. After blocking, the nitrocellulose membranes were incubated overnight at 4 °C with the appropriate primary antibody diluted in OB. Following incubation with the primary antibody, membranes were washed 4 times for 5 min with 0.1% Tween-20 in PBS. The primary antibodies used were: mouse monoclonal anti-penta-His antibody (Qiagen, Inc., Hilden, Germany) (1:3000 in OB/PBS) or mouse monoclonal anti-6xHis antibody (Rockland Immunochemicals, Inc., Limerick, PA, USA) (1:1000 in OB/PBS + 0.2 % Tween), used for the detection of Lam2(482–1259)-(His)_6_ and derived fragments, rabbit polyclonal anti-GST antibody (Santa Cruz Biotechnology, Dallas, TX, USA), used for detection of GST-Laf1 (1:2 × 10^4^), and rabbit polyclonal anti-HA antibody (BioLegend, Inc., San Diego, CA, USA), used to detect Ypk1-(His)_6_-HA (1:1 × 10^4^). Fluorescently tagged proteins were analyzed with rabbit polyclonal anti-tRFP (Evrogen) (1:1000) to detect Laf1-mKate, rabbit monoclonal anti-GFP (D5.1) (Cell Signaling 2956S) (1:500) to detect GFP tagged proteins, and mouse anti-mNeonGreen (mNG) monoclonal antibody (Chromotek 32F6) (1:1000) to detect Lam2-mNG and Laf1-mNG. Pgk1 detected with rabbit polyclonal anti-Pgk1 antibodies (1:2 × 10^4^) prepared in this laboratory [53] were used as the control for equivalent loading of yeast cell extracts.

For detection of the filter-bound primary antibodies, the following infrared dye-labeled secondary antibodies, as appropriate, were used (1:1 × 10^4^ in a 50:50 mixture of OB and 0.1% Tween-20 in PBS and also containing 0.002% SDS): goat anti-mouse immunoglobulin, or goat anti-rabbit IgG (Biotium, Inc., Fremont, CA). After incubation for 1 h at room temperature, the membranes were washed 4 times for 5 min with 0.1% Tween-20 in PBS, followed by a single wash in PBS. The resulting washed filters were visualized and analyzed using an Odyssey™ CLx infrared imaging system (Li-Cor Biosciences).

### 2.8. Fluorescence Microscopy

Cells were grown at 30 °C to mid-exponential phase in synthetic medium containing the appropriate amino acids and bases for plasmid maintenance. Live cells were immobilized between a slide and a coverslip and visualized at room temperature. Fluorescence microscopy for GFP-Lam2 localization assays was performed using an Elyra PS.1 structured illumination (SIM) microscope (Carl Zeiss AG, Jena, Germany) equipped with a 100x PlanApo 1.46NA TIRF objective, a main focus drive of the AxioObserver Z1 Stand, a WSB PiezoDrive 08, controlled by Zen, and images were recorded using a 512 × 512 (100 × 100 nm pixel size) EM-CCD camera (Andor Technology, South Windsor, CT, USA). Images were processed using Fiji and Photoshop (Adobe Systems, Inc., San Jose, CA, USA) [54]. To assess co-localization of GFP-Lam2 and Laf1-mKate, yeast were grown to mid-exponential phase and viewed directly under an epifluorescence microscope (model BH-2; Olympus America, Inc., Center Valley, PA) using an 100X objective equipped with appropriate band-pass filters (Chroma Technology, Corp., Inc. Tuscon, AZ, USA) and processed with µManager [55] and Photoshop (Adobe Systems, Inc., San Jose, CA, USA).

## 3. Results

### 3.1. Localization of β-Propeller Proteins Laf1 and Dgr2 at ER-PM CSs Requires Lam2 and Lam4

It has been demonstrated previously that Lam2 and Lam4 are located at ER-PM CSs, where they act in retrograde transport of ergosterol from the PM to the ER [15,37]. Using immuno-enrichment of tagged LAM/LTC gene products from whole-cell extracts and mass spectrometry of the resulting complexes, it was deduced by others that Lam2 and Lam4 associate with two, as yet poorly characterized, but paralogous proteins, Laf1/Ymr102c and Dgr2/Ykl121w [38]. Based on structure prediction at the Saccharomyces Genome Database and our own sequence analysis (Appendix A), it appears that Laf1 and Dgr2 are both ten-bladed β-propeller proteins.

If these two novel proteins associate physically with Lam2 and Lam4 in vivo, then Laf1 and Dgr2 should also localize at ER-PM CSs. Using fluorescence microscopy, we verified (Figure 1A) that Laf1-mKate co-localizes with GFP-Lam2 in discrete puncta at the PM, and these sites are distinct from eisosomes (marked with the BAR domain protein Pil1-BFP) [38].

The vast majority of ER-PM CSs in *S. cerevisiae* require for their assembly a set of six proteins from three families: two VAMP-associated proteins (VAPs), Scs2 and Scs22, a TMEM16-like protein, Ist2, and three Extended-Synaptotagmin (E-Syt) orthologs, also called tricalbins, Tcb1, Tcb2, and Tcb3 [48]. All six proteins are integral to the ER membrane and bind to the PM by interacting with lipid head groups or other proteins [56]. Removal of all six proteins (a hextuple mutant designated “*tether∆*”) is required to largely abrogate formation of ER-PM CSs [48]. In the *tether∆* mutant, the number of puncta containing either Lam2-mNG or Laf1-mNG was reduced, but not abolished, and the remaining puncta were brighter and less uniformly distributed around the cell periphery than in the purportedly isogenic control cells (Figure 1B), suggesting that these sterol transport proteins may have both the capacity to self-associate and to serve a bridging function that contributes to formation or stability of ER-PM CSs [15,48,57].

Given that both Laf1 and Dgr2 co-localize with Lam2 (and Lam4) at ER-PM CSs, yet neither Laf1 nor Dgr2 are integral membrane proteins, we examined whether the association of Laf1 and Dgr2 with Lam2 and/or Lam4 was necessary for their recruitment to those sites. We found that in cells lacking Lam2 alone, the number of cortical puncta containing Laf1-mNG was drastically reduced, whereas the absence of Lam4 had a much milder effect, and in cells lacking both Lam2 and Lam4, almost no Laf1-mNG was detectable at the PM (Figure 1C, upper panel). The dramatic loss of Laf1-mNG at the PM could not be attributed to any marked reduction in total Laf1-mNG because the amount of Laf1-mNG remained very similar in all four backgrounds (Figure 1C, lower panels).

The exact same results were obtained in cells expressing Laf1-mKate instead of Laf1-mNG (Appendix A). In addition, the accompanying immunoblot analysis of Laf1-mKate expression revealed more clearly that, in WT cells, Laf1-mKate displays slower migrating species that are absent in cells lacking either Lam2 or Lam4 (or both) (Appendix A). Treatment with calf intestinal phosphatase collapsed these slower mobility species (Appendix A), confirming that they arise from phosphorylation (and not some other post-translational modification). This mobility shift was also eliminated in *ypk1-as ypk2∆* cells (Appendix A), which have reduced total Ypk activity because they completely lack Ypk2 and because, as we have documented previously [36,51], the analog-sensitive Ypk1-as mutant has, at best, a specific activity only ~50% of WT, even in the absence of the inhibitor 3MB-PP1. Consistent with this observation, Laf1 contains one consensus Ypk1 phospho-acceptor site (**RxRxxSΦ,** where Φ indicates a preference for a hydrophobic or uncharged residue), namely **R**R**R**FN**S**^709^S, and Dgr2 has two (**R**S**R**HS**S**^49^I and **R**C**R**LW**S**^349^I). To further explore this relationship, we performed an in vitro kinase assay, in which a Laf1(684–834) fragment containing its Ser709 site or a Dgr2(1–128) fragment containing its Ser49 site, or corresponding mutants Laf1(684–834; S709A S710A) and Dgr2(1–128; S48A S49A) in which each of the potential phospho-acceptor residues was converted to Ala, were incubated with [γ-^32^P]ATP and a preparation of Ypk1-as-(His)_6_-HA (Appendix A). We found that both fragments were phosphorylated (although Laf1 was a much better substrate) and that phosphorylation of both fragments was almost completely abolished by the specific Ser-to-Ala mutations, even though both fragments contain numerous other Ser and Thr residues (Laf1: 19S, 10T; Dgr2: 14S, 8T). Despite this evidence for specificity, incorporation into Laf1(684–834) and Dgr2(1–128) catalyzed by the preparation of Ypk1-as-(His)_6_-HA was not blocked at a concentration of the inhibitor 3MB-PP1 that was quite efficacious in reducing incorporation into a control protein, Orm1, a well-documented substrate for Ypk1 both in vivo and in vitro [50,58]. Therefore, in addition to Ypk1, some other as yet unidentified protein kinase in the Ypk1-as-(His)_6_-HA preparation is capable of phosphorylating these proteins at these specific sites. In this regard, it has been reported recently that some targets that contain sequences that harbor within them the Ypk1 consensus can also be phosphorylated by other AGC family protein kinases, including PKA and Sch9 [59]. We have not yet tested the Ypk1-as-(His)_6_-HA preparation for the presence of either of these potential contaminants.

As we observed for Laf1-mNG and Laf1-mKate, we likewise found that in cells lacking Lam2 alone, the number of cortical puncta containing Dgr2-mKate was drastically reduced, whereas the absence of Lam4 had a milder effect, and, in cells lacking both Lam2 and Lam4, almost no Dgr2-mKate was detectable at the PM (Figure 1D, upper panel). Again, the dramatic loss of Dgr2-mKate at the PM could not be attributed to any marked reduction in total Dgr2-mKate because the amount of Dgr2-mKate remained very similar in all four backgrounds (Figure 1D, lower panels). Also, like Laf1, based on its electrophoretic isoforms, Dgr2 appears to be a phosphoprotein.

We also examined the converse relationship. We found that localization of GFP-Lam2 to ER-PM CSs was not detectably diminished in cells lacking both Laf1 and Dgr2 (Figure 1E). Given that Lam2 is anchored in the ER membrane via its C-terminal transmembrane domain, these results establish that it is Lam2 and Lam4 that are required to recruit Laf1 and Dgr2 to ER-PM CSs, and association with Laf1 and Dgr2 is not required for Lam2 or Lam4 to reach and remain at ER-PM CSs.

### 3.2. Laf1 Is Required for Efficient Retrograde Removal of PM Ergosterol

Having established that Lam2 and Lam4 are required to recruit Laf1 and Dgr2 to ER-PM CSs and that all four proteins co-localize there, we then tested whether Laf1 and/or Dgr2 had, like Lam2 and Lam4, a role in retrograde sterol transport. Due to their deficiency in transferring ergosterol from the PM back to the ER, cells lacking Lam2 and Lam4 have a higher concentration of accessible ergosterol in their PM than WT cells. This defect makes *lam2∆ lam4∆* cells hypersensitive to the anti-fungal agent amphotericin B (AmB) [15,37] because this compound kills cells by sequestering ergosterol from the PM [60]. Similarly, we found that, in cells expressing GFP-Lam2, absence of Laf1 was sufficient to make the cells more sensitive to the toxic effect of AmB compared to the otherwise isogenic *LAF1^+^* (control) cells (Figure 2A), whereas absence of Dgr2 alone had no discernible effect and did not further enhance the AmB sensitivity of Laf1-deficient cells (Figure 2A). Thus, under normal conditions, Laf1, and not Dgr2, plays the major role in the events required for retrograde sterol transport.

The most straightforward explanation for this distinction is that Laf1 is a substantially more abundant protein than Dgr2 based on two criteria. First, estimates of their relative levels calculated on the basis of all available proteomic studies of *S. cerevisiae* [61] show that Laf1 is three times more abundant than Dgr2 (average number of molecules per cell of 1580 versus average number of molecules per cell of 540) [61]. Second, in our hands, when tagged with the same epitope (mKate) and analyzed by immunoblotting with the same antibody (anti-tRFP), we find that Laf1 is indeed expressed at a much higher level than Dgr2 (Figure 2B).

Most significantly, when assessed in otherwise WT cells, the degree of AmB sensitivity conferred by loss of Laf1 was significantly less severe than that conferred by loss of Lam2 and Lam4 (Figure 2C). It is telling, however, that when a *laf1∆* mutation was installed in *lam2∆ lam4∆* double mutant cells, we did not observe any enhancement of their AmB sensitivity (Figure 2C). This lack of an additive effect is genetic evidence that Laf1 acts together functionally with Lam2 and Lam4 in the same sterol transport process, consistent with the co-localization of these proteins.

### 3.3. Laf1 Physically Associates with Lam2 and the PH Domain of Lam2 Is Not Required for Its Interaction with Laf1

To determine whether the co-localization and shared function of Lam2 and Lam4 with Laf1 and Dgr2 are due to their direct physical interaction, we expressed GST-tagged versions of Laf1 and Dgr2 and (His)_6_-tagged soluble fragments of Lam2 and Lam4 in *E. coli* and examined the ability of the resulting purified recombinant proteins to associate with each other in vitro using a standard pull-down assay. The Lam2 and Lam4 constructs comprised most of their shared cytoplasmic portions, including their PH and two StARkin domains, but excluded their C-terminal transmembrane ER anchor (Figure 3A, upper panel).

The largest Lam2 construct, Lam2(482–1259), contains two of its three documented Ypk1 phosphorylation sites (T518 and T1237) and the Lam4 construct, Lam4(371–1177), contains its two documented Ypk1 phosphorylation sites (S401 and S667) [37]. When incubated in solution and then affinity-captured on glutathione-agarose beads, GST-Laf1 or GST-Dgr2 were able to bind both the Lam2(482–1259) and Lam4(371–1177) fragments, whereas a huge excess of GST alone did not detectably bind Lam2(482–1259) and exhibited only weak background level binding of Lam4(371-1177) (Figure 3A, lower panel). Because Lam2 is more abundant than Lam4 (average number of molecules per cell of 1890 versus average number of molecules per cell of 890) [61] and because we previously demonstrated that loss of Lam2 has a more dramatic effect in impeding retrograde sterol transport in vivo than does loss of Lam4 [37], and because Laf1 is much more abundant than Dgr2, as we documented above, we focused our attention on defining the Laf1-Lam2 interaction in somewhat greater detail.

Compared to Lam2(482-1259)-(His)_6_, an N-terminally truncated derivative that still retained the PH domain, Lam2(633-1259)-(His)_6_ (Figure 3B, upper panel), bound to GST-Laf1 much more weakly, even though the input amount of both fragments was equivalent (Figure 3B, lower panel). Most strikingly, an even shorter fragment from which the PH was removed, Lam2(849–1259)-(His)_6_ (as well as a proteolytic product) bound markedly more efficiently to GST-Laf1 than Lam2(482–1259)-(His)_6_. These results demonstrate that it is the tandem StARkin domains of Lam2 that mediate its association with Laf1, and that the PH domain may have an inhibitory role in regulating the interaction of Lam2 with Laf1, which may be further modulated by the upstream sequence that contains the one Ypk1 phosphorylation site that is shared between Lam2 and Lam4. 

To further verify the conclusion that the PH domain is inhibitory to Lam2-Laf1 interaction, we generated an internal deletion (∆644–760) in the Lam2(482–1259)-(His)_6_ fragment, which excised just the PH domain (Figure 3C, upper panel). We found that, just like the Lam2(849–1259) fragment (from which the PH domain was removed by N-terminal truncation), the Lam2(482–1259; ∆644–760)-(His)_6_ fragment (from which the PH domain was removed by internal deletion) bound to GST-Laf1 much more robustly than Lam2(482–1259)-(His)_6_ itself (Figure 3C, lower panel). Thus, the PH is certainly dispensable for Lam2-Laf1 association and, in fact, hinders their interaction. Also, neither mutation to Ala of both of the potential Ypk1 phospho-acceptor residues in GST-Laf1 (S709S S710A) nor mutation to Ala of the conserved Ypk1 phospho-acceptor site and its adjacent reside (T518A L519A) in the Lam2(482-1259)-(His)_6_ fragment, cause any detectable change in the Lam2-Laf1 interaction, as expected for recombinant fragments generated in *E. coli* where phosphorylation is absent.

### 3.4. Ypk1-Mediated Phosphorylation of Lam2 Disrupts Its Association with GST-Laf1

As one approach to examine whether phosphorylation of Lam2 by Ypk1 and/or Ypk2 influences the stability or localization of Laf1, we examined the status of Laf1-mKate expressed from its endogenous chromosomal locus in cells expressing WT Lam2, or Lam2, in which each of the three Ypk1 phospho-acceptor residues (as well as the immediately downstream residue at each site) were all mutated either to Ala (to prevent phosphorylation by Ypk1 and Ypk2) (abbreviated A) or to Glu (to mimic full phosphorylation by Ypk1 and Ypk2) (abbreviated E) (Figure 4A, upper panel). These Lam2 constructs were GFP-tagged and integrated at the LAM2 locus in a strain carrying a lam4∆ mutation, so that these Lam2 derivatives were the only sterol transporter of its class present. When examined by immunoblotting, GFP-Lam2, GFP-Lam2(A), and GFP-Lam2(E) were all expressed at an equivalent level and the level of Laf1-mKate in each of these backgrounds was unaffected (Figure 4A, lower panel). Thus, the state of Lam2 phosphorylation at its Ypk1 sites does not influence Laf1 stability.

Because both Laf1 and Lam2 and its derived mutants were tagged with fluorescent proteins, we could also examine whether the state of Lam2 phosphorylation at its Ypk1 sites affected either Laf1 or Lam2 localization. It appeared that WT GFP-Lam2, GFP-Lam2(A), and GFP-Lam2(E) all occupied an equivalent number of ER-PM CSs around the cell periphery and, most significantly, that the fraction of those sites to which Laf1-mKate co-localized was quite similar in all three strain backgrounds (Figure 4B). Thus, the state of Lam2 phosphorylation at its Ypk1 sites does not influence Laf1 retention at ER-PM CSs.

Prior work has demonstrated that inhibition of sphingolipid biosynthesis by antibiotics like myriocin [50] or aureobasidin A [62] is the most potent known means to stimulate TORC2-dependent activation of Ypk1 and, in turn, Ypk1-mediated phosphorylation of its substrates. Therefore, we treated the *lam4∆* strain co-expressing Laf1-mKate and WT GFP-Lam2 with myriocin to maximize PM stress-induced stimulation of Ypk1, and then examined GFP-Lam2 and Laf1-mKate localization and levels. We found that myriocin treatment had no dramatic effect on either the localization of GFP-Lam2 at ER-PM CSs or on the fraction of those sites to which Laf1-mKate co-localized (Figure 4C, upper panels). Moreover, after treatment of proteins in cell extracts with calf intestinal phosphatase to collapse all phospho-isoforms, we found that myriocin treatment did not markedly alter the total levels of either GFP-Lam2 or Laf1-mKate (Figure 4C, lower panels). Importantly, however, in the context of our in vitro pull-down assay, we found that a phospho-mimetic variant of Lam2(482-1259)-(His)_6_, Lam2(482-1259; T518E L519E T1237E V1238E)-(His)_6_, bound much more poorly to GST-Laf2 than a corresponding non-phosphorylatable derivative, Lam2(482-1259; T518A L519A T1237A V1238A)-(His)_6_ (Figure 4D), suggesting that even if Laf1 is able to remain stable and in proximity to phosphorylated Lam2 at ER-PM CSs, their ability to physically associate is impaired.

To assess whether actual phosphorylation of Lam2 is just as disruptive to its ability to associate with Laf1 as the phospho-mimetic Glu mutations at the Ypk1 consensus sites in Lam2(482–1259)-(His)_6_, we worked out conditions for authentic in vitro phosphorylation by purified Ypk1-(His)_6_-HA that allowed us to quantitatively convert all of the Lam2(482–1259)-(His)_6_ fragments present to phospho-isoforms (Figure 5A). Satisfyingly, we found that compared to the robust binding of unphosphorylated Lam2(482–1259)-(His)_6_ to GST-Laf1, the binding of the phosphorylated Lam2(482–1259)-(His)_6_ to GST-Laf1 was dramatically reduced (Figure 5B). Thus, the primary, and perhaps sole, effect of Ypk1-mediated phosphorylation of Lam2 is to prevent its ability to engage in an intimate protein–protein interaction with Laf1. The implications of our results are discussed below in the Discussion Section.

## 4. Discussion

The present study was undertaken to understand the mechanism by which Ypk1-mediated phosphorylation of Lam2 and Lam4 inhibits their function in retrograde transport of ergosterol at ER-PM CSs [37]. One clue was buried in a prior observation that, in the compartment occupied by Lam2 and Lam4, there were also two poorly characterized proteins present, Ymr102c and Dgr2 [38]. We found this finding intriguing for several reasons. First, close inspection of their sequences and use of predictive algorithms [63,64] indicated that both Ymr102c and Dgr2 contain ten WD40 repeat motifs. The WD40 repeat, first described in bovine β-transducin [65], is a conserved stretch of ~40 residues, often with Trp (W) (or other aromatics) and Asp (D) (or other charged residues) at its C-terminal end and often with Gly and His present with the sequence near its N-terminal end. A WD40 repeat typically folds into a four-stranded anti-parallel β-sheet that constitutes a blade of a β-propeller fold [39].

Second, proteins with β-propeller domains are very frequently involved in directly mediating protein–protein interactions in both the yeast and human protein interactomes [40,41,42,43,44]. Indeed, as we have demonstrated here, Ymr102 and Dgr2 both co-localize with Lam2 and Lam4 in vivo and, as purified recombinant proteins, bind to Lam2 and Lam4 in vitro in the absence of any other components. Hence, given that we have demonstrated most clearly that Ymr102c is a direct interactor with Lam2, we have recommended to the *Saccharomyces* Genome Database that that the *YMR102c* locus be redesignated the *LAF1* gene for “Lam2/Ltc4-Associated Factor 1.”

Third, most β-propeller proteins have seven blades, but representatives with either a lesser or a greater number of WD40 repeats are known from crystal structures [66], even β-propeller proteins with ten. For example, the ectodomain of neurotensin receptor 3 (NTS3), also known as sortilin, is a ten-bladed β-propeller with a typical toroidal topology that has a large enough central cavity for specific binding of the 13-residue neurotransmitter peptide neurotensin [67]. Thus, with ten predicted WD40 repeats, Laf1 and Dgr2 also both likely adopt a ten-bladed β-propeller topology. Moreover, given the genetic and other evidence we have presented here that Laf1 and Dgr2 have a function in retrograde sterol transport, in conjunction with Lam2 and Lam4, we suspect that the residues lining the central cavity in the Laf1 and Dgr2 ten-bladed β-propellers selectively accommodate ergosterol. Fully consistent with this supposition, using a photoactivatable sterol derivative, Laf1 (Ymr102c) was detected recently as a sterol-binding protein in yeast extracts [68]. The fact that Laf1 is at least three times more abundant than Dgr2 possibly explains why Dgr2 was not detected by the same approach. In this same regard, we documented here that Laf1 interacts with the tandem sterol-binding StARkin domains of Lam2. Taken together, these findings suggest a model in which Laf1 has a direct role in how Lam2 extracts ergosterol from the PM and/or in how Lam2 delivers the sterol back to the ER. Perhaps these two distinct classes of sterol-binding motifs—the StARkin domains in Lam2 and the β-propeller in Laf1—act in a “bucket brigade” fashion to transmit ergosterol from one membrane, then between each other, and then to the other membrane. It will be interesting to determine the relative affinities of these two sterol-binding modalities and their order of function in the retrograde sterol transport process.

Finally, as we demonstrated here, Ypk1-mediated phosphorylation of Lam2 does not affect the stability or localization either of Lam2 itself or of Laf1. The main effect of Lam2 phosphorylation by Ypk1 is to block its protein–protein interaction with Laf1, thereby disrupting the contact between these two proteins. Thus, the most parsimonious interpretation for how Lam2 phosphorylation by Ypk1 inhibits retrograde sterol transport is that it blocks the ability of Lam2 and Laf1 to interact, thereby impeding coordination of their mutual and cooperative functions in transit of ergosterol from the PM to the ER. Because phosphorylation-mediated disruption of Lam2–Laf1 association does not displace Laf1 from Lam2-containing ER-PM CSs, yet Lam2 is essential for Laf1 localization to ER-PM CSs, Laf1 likely makes multiple contacts with Lam2, and only those required for their coordinate interaction in ergosterol transfer are disrupted by phosphorylation. In this same regard, our in vitro analysis of Laf1–Lam2 interaction indicated that the PH domain in Lam2 is quite inhibitory to its association with Laf1, and that sequences immediately N-terminal to the PH seem to alleviate that inhibitory effect. Tellingly, that upstream sequence contains the one Ypk1 phosphorylation site that is conserved between Lam2 and Lam4 [37]. These interrelationships further extend our model for how Lam2–Laf1 function is negatively regulated by Ypk1-mediated phosphorylation. In the absence of phosphorylation, the sequence in Lam2 upstream sequesters the PH domain and prevents it from associating with the tandem StARkin domains, thereby allowing the StARkin domains to associate productively with Laf1. However, when Ypk1 phosphorylates its site in the upstream sequence, the PH domain is displaced and is free to associate with and occlude the StARkin domains, thus blocking their proper mode of association with Laf1, thereby impeding the specific interactions between Lam2 and Laf1 that are required for their joint function in retrograde ergosterol transport. 

## Figures and Tables

**Figure 1 biomolecules-10-01598-f001:**
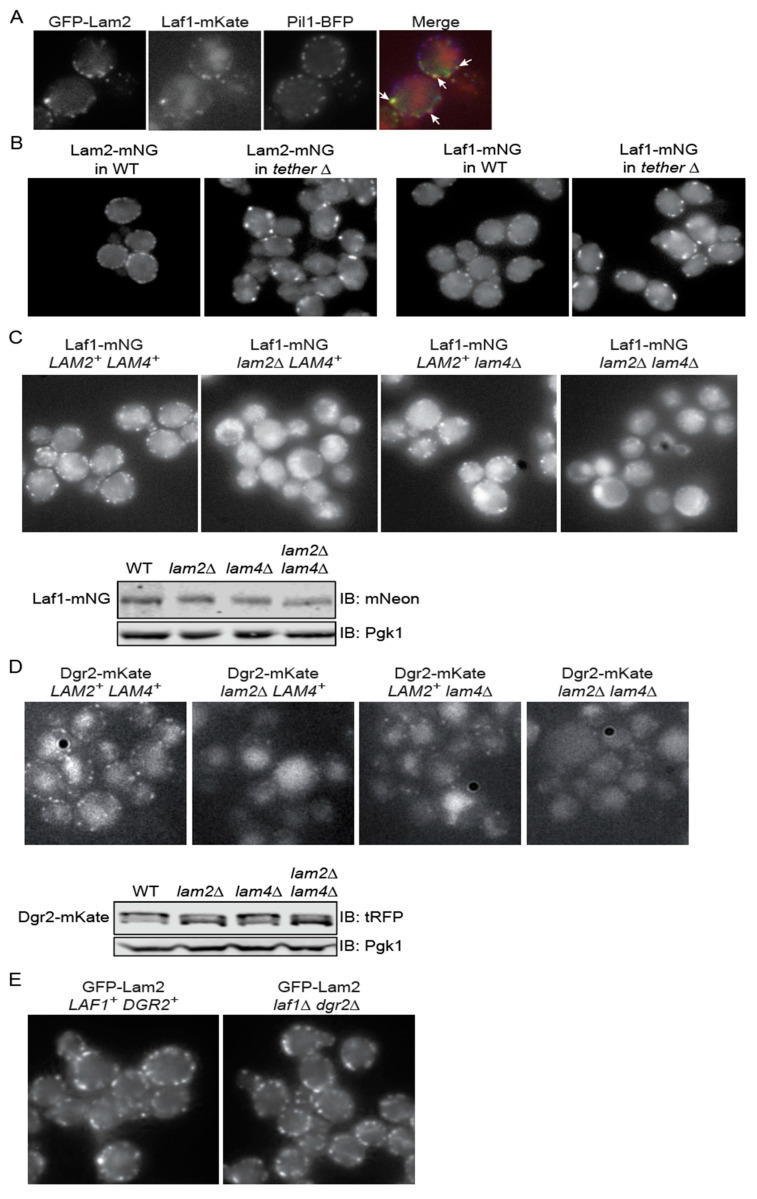
(**A**) Laf1 co-localizes with Lam2 at ER-PM CS contact sites. A strain (yEPS23) co-expressing Laf1-mKate, GFP-Lam2, and Pil1-GFP, each from its respective endogenous chromosomal locus, was grown to mid-exponential phase and viewed directly under an epifluorescence microscope, as described in the Materials and Methods Section. (**B**) Localization of Lam2 and Laf1 to ER-PM CSs is reduced, but not abolished, in the *tether*∆ strain. A wild-type strain (YFR712) and an otherwise isogenic *tether*∆ strain (YFR704) expressing Lam2-mNG (left panels) and the wild-type strain (YFR729) and the otherwise isogenic *tether*∆ strain (YFR705) expressing Laf1-mNG (right panels) were grown to mid-exponential phase in YPD and viewed directly under an epifluorescence microscope, as described in the Materials and Methods Section. (**C**) Localization of Laf1 to ER-PM CSs requires Lam2 and Lam4. Upper panels, Strains expressing Laf1-mNG in a wild-type (YFR713), *lam2∆* (YFR714), *lam4∆* (YFR715), or *lam2∆ lam4∆* (YFR720) background, were grown to mid-exponential phase and viewed directly under an epifluorescence microscope. Lower panels, samples of equivalent amounts (as judged by the Pgk1 loading control) of extracts of the same cells shown in the upper panel were resolved by SDS-PAGE and analyzed by immunoblotting, as described in the Materials and Methods Section. (**D**) Localization of Dgr2 to ER-PM CSs requires Lam2 and Lam4. Upper panels, Strains expressing Dgr2-mKate in a wild-type (YFR657-B), *lam2∆* (YFR658-B), *lam4∆* (YFR659-B), or *lam2∆ lam4∆* (YFR650-B) background, were grown to mid-exponential phase and viewed directly under an epifluorescence microscope. Lower panels, samples of equivalent amounts (as judged by the Pgk1 loading control) of extracts of the same cells shown in the upper panel were resolved by SDS-PAGE and analyzed by immunoblotting, as described in the Materials and Methods Section. (**E**) Lam2 localizes to ER-PM CSs in the absence of Laf1 and Dgr2. A WT strain (YFR512-A) and an otherwise isogenic *laf1∆ dgr2∆* derivative (YFR613-A), each expressing GFP-Lam2, were grown to mid-exponential phase and viewed using an Elyra PS.1 structured illumination (SIM) fluorescence microscope.

**Figure 2 biomolecules-10-01598-f002:**
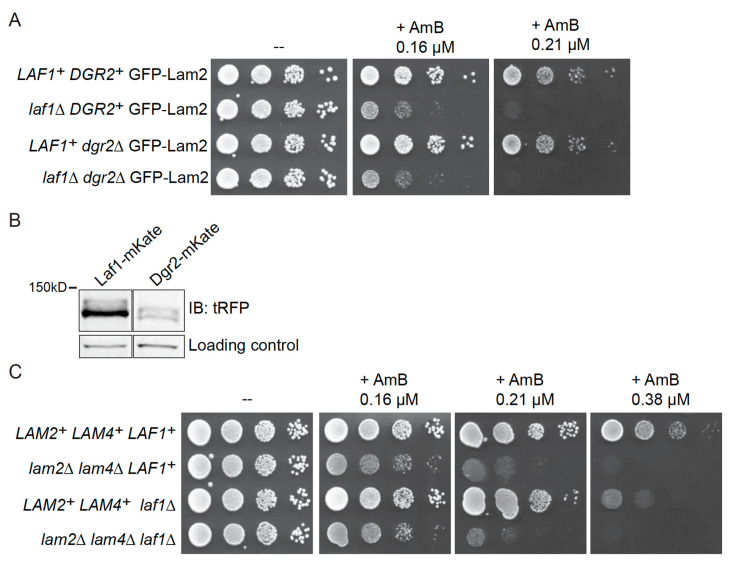
(**A**) Laf1 has a function in retrograde sterol transport. Serial 10-fold dilutions of WT GFP-Lam2 (YFR512-A), *laf1∆* GFP-Lam2 (YFR603), *dgr2∆* GFP-Lam2 (YFR607), and *laf1∆ dgr2∆* GFP-Lam2 (YFR613-A) were spotted on plates lacking (-) or containing AmB at the indicated concentrations. The plates were scanned after 2 days of growth at 30 °C. (**B**) Laf1 is more abundant than Dgr2. Extracts of cells expressing, as indicated, either Laf1-mKate (YFR635) or Dgr2-mKate (YFR657-A), each from their endogenous chromosomal locus, were resolved by SDS-PAGE and analyzed by immunoblotting, as described in the Materials and Methods Section. (**C**) Laf1 functions in the same retrograde sterol transport pathway as Lam2 and Lam4. Serial 10-fold dilutions of WT (BY4741), *lam2∆ lam4∆* (YFR513), *laf1∆* (*ymr102c∆*), and *lam2∆ lam4∆ laf1∆* (YFR734-A) were spotted on plates lacking (-) or containing AmB at the indicated concentrations. The plates were scanned after 2 days of growth at 30 °C.

**Figure 3 biomolecules-10-01598-f003:**
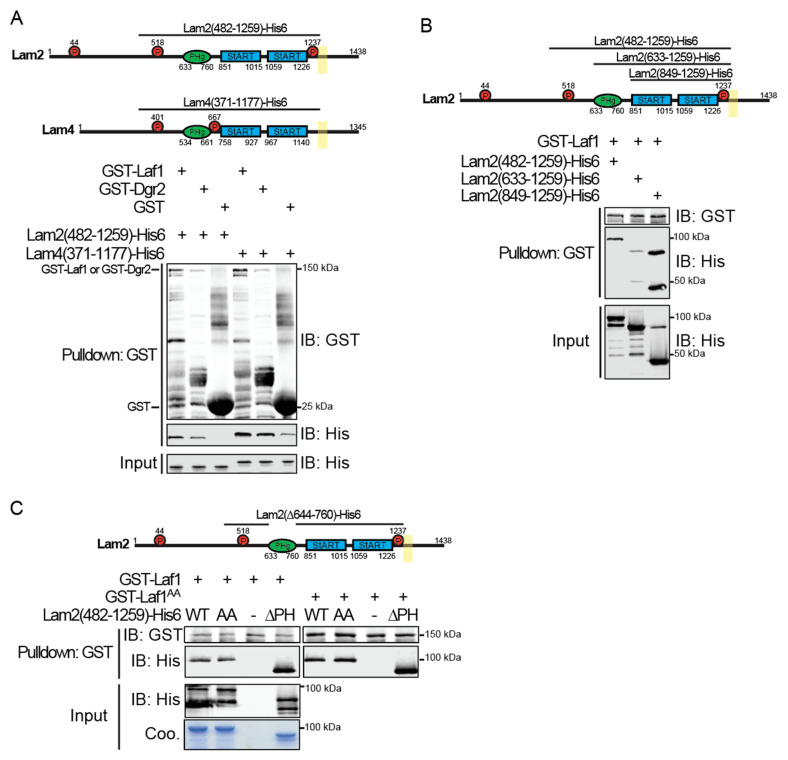
(**A**) Immobilized Laf1 and Dgr2 bind soluble fragments of Lam2 and Lam4 in vitro. Upper panel, structural motifs present in the Lam2(482–1259)-(His)_6_ (pFR391) and Lam4(371–1177)-(His)_6_ (pFR407) constructs. Lower panel, GST-Laf1 (pMT2), GST-Dgr2 (pFR404), and GST alone were expressed in *E. coli* from the vector pGEX4T-1 and purified. Samples were incubated in solution with, as indicated, either Lam2(482–1259)-(His)_6_ or Lam4(371–1177)-(His)_6_ for 1 h, and then the resulting complexes captured on glutathione-agarose beads, as described in the Materials and Methods Section. After washing, bound proteins were eluted, resolved by SDS-PAGE, and analyzed by immunoblotting. (**B**) The PH domain of Lam2 impedes its binding to Laf1. Upper panel, structural motifs present in the Lam2(482-1259)-(His)_6_ (pFR391), Lam2(633-1259)-(His)_6_ (pFR408), and Lam2(849-1259)-(His)_6_ (pFR409) constructs. Lower panel, GST-Laf1 (pMT2) was purified from *E. coli*, incubated with the three different Lam2 fragments indicated for 1 h, then the resulting complexes were affinity-captured on glutathione-agarose beads, as described in the Materials and Methods Section. After washing, bound proteins were eluted, resolved by SDS-PAGE, and analyzed by immunoblotting. (**C**) The PH domain in Lam2 is dispensable for its interaction with Laf1. Upper panel, structural motifs present in the Lam2(482-1259)-(His)_6_ (pFR391), Lam2(482-1259; T518A L519A)-(His)_6_ (abbreviated AA) (pFR392), and Lam2(482-1259; ∆644-760-(His)_6_ (abbreviated ∆PH) (pFR393) constructs. Lower panel, GST-Laf1 (pMT2) and GST-Laf1(S709A S710A) (abbreviated AA) (pFR396) were purified from *E. coli*, incubated with the three different Lam2 fragments indicated for 1 h, then the resulting complexes were affinity-captured on glutathione-agarose beads, as described in the Materials and Methods Section. After washing, bound proteins were eluted, resolved by SDS-PAGE, and analyzed by immunoblotting.

**Figure 4 biomolecules-10-01598-f004:**
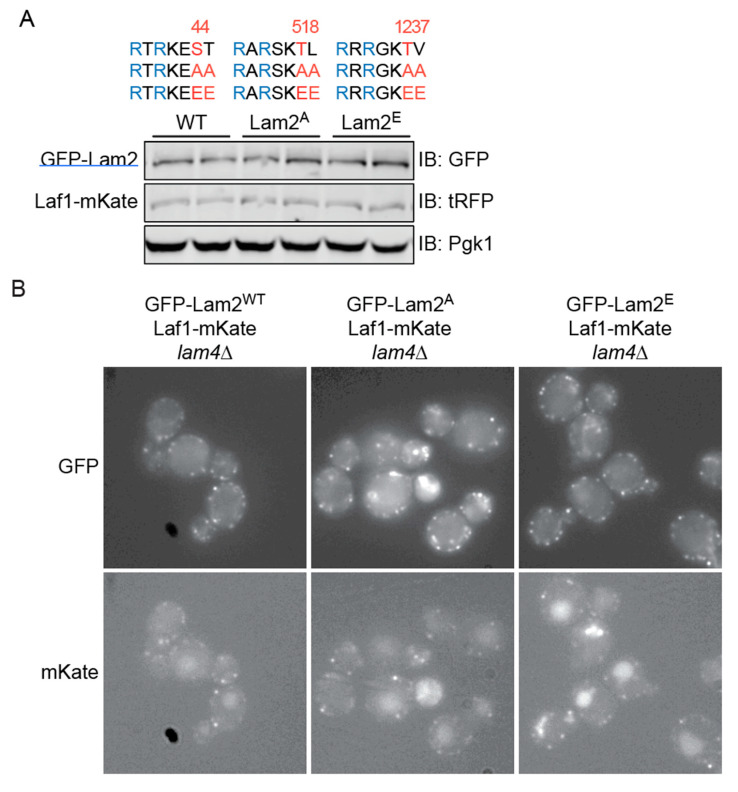
(**A**) State of Lam2 phosphorylation does not affect Laf1 stability. Upper panel, the three documented Ypk1 phosphorylation sites in Lam2, and the hextuple “A” mutant (S44A T45A T518A L519A T1237A V1238A) and the hextuple “E” mutant (S44E T45E T518E L519E T1237E V1238E). Lower panel, extracts of *lam4∆* cells co-expressing Laf1-mKate with either GFP-Lam2^WT^ (YFR728-A and YFR728-B), GFP-Lam2^A^ (YFR721-A and YFR721-B), and GFP-Lam2^E^ (YFR722-A and YFR722-B) were resolved by SDS-PAGE and analyzed by immunoblotting. (**B**) Phosphorylation state of Lam2 does not affect Laf1 retention at ER-PM CSs. The *lam4∆* strains co-expressing Laf1-mKate and, as indicated, WT GFP-Lam2 (YFR728-B), GFP-Lam2^A^ (YFR721-B), or GFP-Lam2^E^ (YFR722-B), were grown to mid-exponential phase in SCD-Trp and viewed under an epifluorescence microscope. (**C**) Stimulation of TORC2-dependent Ypk1 activation does not affect either Lam2 or Laf1 localization or level. Upper panels, *lam4∆* cells co-expressing WT GFP-Lam2 and Laf1-mKate (YFR633-A) cells were grown to mid-exponential phase, then either mock-treated (-) or treated (+Myr) with myriocin (1.25 µM) for 2 h before viewing under an epifluorescence microscope. Lower panels, extracts of another *lam4∆* strain co-expressing WT GFP-Lam2 and Laf1-mKate (YFR726-A) were prepared, treated with calf intestinal phosphatase, resolved by SDS-PAGE, and analyzed by immunoblotting. (**D**) Phospho-mimetic mutation in Lam2(482–1259)-(His)_6_ reduce its affinity for GST-Laf1. Two Lam2(482–1259)-(His)_6_ mutants, Lam2(482-1259; T518A L519A T1237A V1238A)-(His)_6_ (pFR411) and Lam2(482–1259; T518E L519E T1237E V1238E)-(His)_6_ (pFR413), were purified from *E. coli*, incubated with purified GST-Laf1 for 1 h, then the resulting complexes were affinity-captured on glutathione-agarose beads, as described in the Materials and Methods Section. After washing, bound proteins were eluted, resolved by SDS-PAGE, and analyzed by immunoblotting.

**Figure 5 biomolecules-10-01598-f005:**
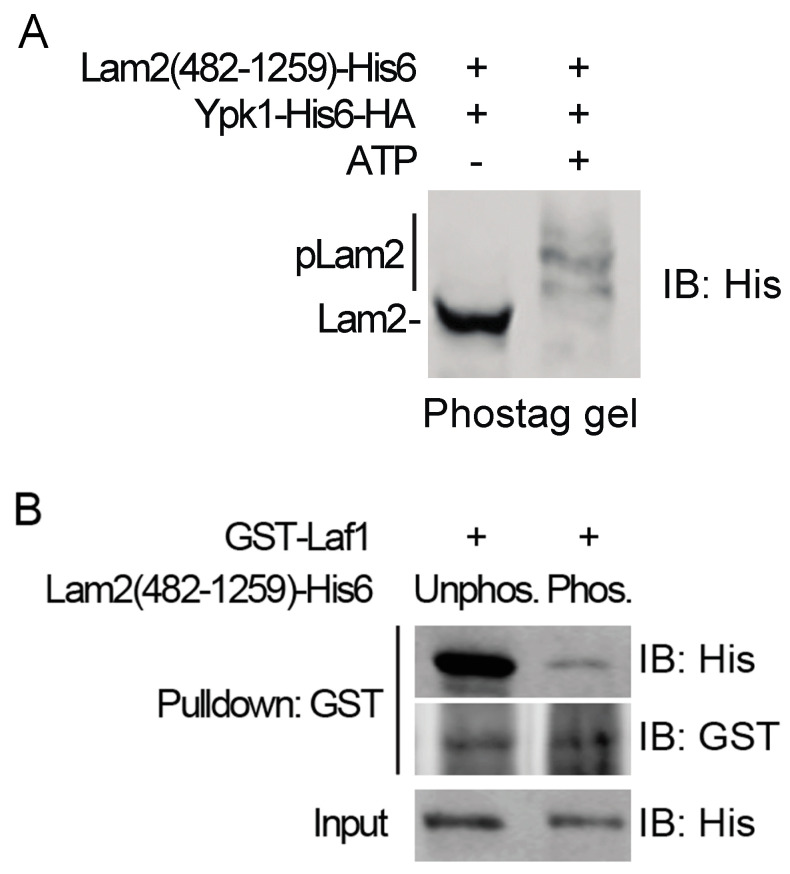
Phosphorylation of Lam2 blocks its interaction with Laf1. (**A**) GST-Lam2(482-1259)-(His)_6_ (pFR391) purified from *E. coli* was incubated with Ypk1-(His)_6_-HA purified, as described in the Materials and Methods Section, from *S. cerevisiae* strain CGA84 containing plasmid BG1805 expressing Ypk1-(His)_6_-HA-3C-ZZ, in the absence (-) or presence (+) of excess Mg-ATP. The products were then resolved by PhosTag™ SDS-PAGE (8% acrylamide gel, 35 µM PhosTag reagent), and analyzed by immunoblotting. (**B**) Purified recombinant GST-Laf1 was incubated for 1 h with the unphosphorylated and phosphorylated Lam2(482–1259)-(His)_6_, prepared as shown in (A), for 1 h, then the resulting complexes were affinity-captured on glutathione-agarose beads. After washing, bound proteins were eluted, resolved by SDS-PAGE, and analyzed by immunoblotting.

**Table 1 biomolecules-10-01598-t001:** *S. cerevisiae* strains used in this study.

Strain	Genotype	Source/Reference
BY4741	*MAT* **a** *his3∆1 leu2∆0 met15∆0 ura3∆0*	Research Genetics, Inc. (Huntsville, AL, USA)
BY4742	*MAT his3∆1 leu2∆0 lys2∆0 ura3∆0*	Research Genetics, Inc.
YFR651-A	BY4741 Laf1-mKate::Sp*HIS5*	This study
YFR652-B	BY4742 Laf1-mKate::Sp*HIS5 lam2∆::*HygMX	This study
YFR653-B	BY4742 Laf1-mKate::Sp*HIS5 lam4∆::*KanMX	This study
YFR654-B	BY4742 Laf1-mKate::Sp*HIS5 lam2∆::*HygMX *lam4∆::* KanMX	This study
YFR660	BY4741 Ypk1(L424A)::*ura3- ypk2*∆::KanMX Laf1-mKate::Sp*HIS5*	This study
YFR657-A	BY4741 Dgr2-mKate:: Sp*HIS5*	This study
YFR657-B	BY4742 Dgr2-mKate:: Sp*HIS5*	This study
YFR658-B	BY4742 Dgr2-mKate:: Sp*HIS5 lam2∆::*HygMX	This study
YFR659-B	BY4742 Dgr2-mKate:: Sp*HIS5 lam4∆::*KanMX	This study
YFR650-B	BY4742 Dgr2-mKate:: Sp*HIS5 lam2∆::*HygMX *lam4∆::* KanMX	This study
YFR512-A	BY4741 GFP-Lam2::*URA3*	[37]
YFR613-A	BY4742 GFP-Lam2::*URA3 laf1∆::*KanMX *dgr2∆*::KanMX	This study
YFR513	BY4741 *lam2∆*:: HygMX *lam4∆*::KanMX	[37]
YFR633-A	BY4741 GFP-Lam2::*URA3* Laf1-mKate::Sp*HIS5*	This study
YFR726-A	BY4742 GFP-Lam2::*URA3* Laf1-mKate::Sp*HIS5*	This study
YFR635	BY4742 Laf1-mKate::Sp*HIS5*	This study
YFR728	BY4742 Laf1-mKate GFP-Lam2::*URA3 lam4∆*::KanMX	This study
YFR721	BY4742 Laf1-mKate::SpHis5 GFP-Lam2(S44A T45A T518A L519A T1237A V1238A)::*URA3 lam4∆*::KanMX	This study
YFR722	BY4741 Laf1-mKate::SpHis5 GFP-Lam2(S44E T45E T518E L519E T1237E V1238E)::*URA3 lam4∆*::KanMX	This study
yEPS23	BY4741 GFP-Lam2::*URA3* Laf1-mKate::Sp*HIS5* Pil1-BFP::KanMX	This study
YFR713	BY4741 Laf1-mNG::*LEU2*	This study
YFR714	BY4741 Laf1-mNG::*LEU2 lam2∆::* HygMX	This study
YFR715	BY4742 Laf1-mNG::*LEU2 lam4∆::* KanMX	This study
YFR720	BY4742 Laf1-mNG::*LEU2 lam2∆::*HygMX *lam4∆::*KanMX	This study
*ymr102c∆*	BY4742 *laf1∆*::KanMX	Research Genetics, Inc.
YFR734-A	*laf1∆*::KanMX *lam2∆*::HygMX *lam4∆*::KanMX	This study
YFR603	BY4741 GFP-Lam2::*URA3 laf1∆*::KanMX	This study
YFR607	BY4741 GFP-Lam2::*URA3 dgr2∆*::KanMX	This study
CGA84	*MAT***a***leu2∆1*::GEV::NATMX *pep4∆::HIS3 prb1∆1.6R ura3-52 trp1-1 lys2-801a leu2∆1 his3∆200 can1 GAL*	[46]
SEY6210	*MAT leu2-3,112 ura3-52 his3∆200 trp1∆901 suc2∆9 lys2*-801*a GAL*	[47]
YFR712	SEY6210 Lam2-mNG::*LEU2*	This study
ANDY198	SEY6210 *ist2*∆::hisMX6 *scs2*∆::*TRP1 scs22*∆::hisMX6 *tcb1*∆::kanMX6 *tcb2*∆::kanMX6 *tcb3*∆::hisMX6	[48]
YFR704	ANDY198 Lam2-mNG::*LEU2*	This study
YFR729	SEY6210 Laf1-mNG::*LEU2*	This study
YFR705	ANDY198 Laf1-mNG::*LEU2*	This study

**Table 2 biomolecules-10-01598-t002:** Plasmids used in this study.

Plasmid	Description	Source/Reference
pGEX4T-1	GST tag, bacterial expression vector	GE Healthcare, Inc.
pFR398	pGEX4T-1 Laf1(684–834)	This study
pFR402	pGEX4T-1 Laf1(684–834) S709S S710A	This study
pFR399	pGEX4T-1 Dgr2(1–128)	This study
pFR403	pGEX4T-1 Dgr2(1–128) S47A S48A	This study
pFR203	pGEX4T-1 Orm1(1–85)	[50]
pMT2	pGEX4T-1 Laf1	This study
pFR396	pGEX4T-1 Laf1(S708A S709A)	This study
pFR404	pGEX4T-1 Dgr2	This study
pKM263	GST tag, bacterial expression vector	
pFR391	pKM263-Lam2(482–1259)-His6	This study
pFR392	pKM263-Lam2(482–1259) T518A T1237A-His6	This study
pFR393	pKM263-Lam2(482–1259)∆PH(644–760)-His6	This study
pFR407	pKM263-Lam4(371–1177)-His6	This study
pFR408	pKM263-Lam2(633–1259)-His6	This study
pFR409	pKM263-Lam2(849–1259)-His6	This study
pGEX6P-1	GST tag, bacterial expression vector	GE Healthcare, Inc.
pFR377	pGEX6P-1 Lam2(482–1259)	This study
pFR411	pGEX6P-1 Lam2(482–1259) T518A L519A T1237A V1238A-His6	This study
pFR413	pGEX6P-1 Lam2(482–1259) T518E L519E T1237E V1238E-His6	This study
BG1805	C-terminal His_6_-HA-3C-ZZ tag, yeast expression vector	GE Healthcare, Inc.
pJT4317	BG1805 Ypk1-His_6_-HA-3C-ZZ	GE Healthcare, Inc.

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
