# Peer review of "TORC2-Dependent Ypk1-Mediated Phosphorylation of Lam2/Ltc4 Disrupts Its Association with the β-Propeller Protein Laf1 at Endoplasmic Reticulum-Plasma Membrane Contact Sites in the Yeast Saccharomyces cerevisiae"

_biomolecules, 2020, doi:10.3390/biom10121598_

Round 1
Reviewer 1 Report
The manuscript by M. Topolska et al. is an interesting and thorough study of the regulation of two sterol transport proteins important for maintaining the lipid composition of the PM. In eukaryotes, each cellular compartment has a unique lipid composition, maintained largely by lipid transfer at membrane contact sites. Two ER transmembrane proteins, Lam2 and Lam4, concentrate at ER-PM contact sites (CS) where they transfer sterol from its site of synthesis in the ER to the PM. The authors had previously identified Lam2 and Lam4 as direct targets of the Ypk1 protein kinase, a key regulator in the TORC2 pathway. Here they show that Ypk1 phosphorylation inhibits sterol transfer by Lam2 and Lam4, through regulation of Lam2/Lam4 interaction with two protein partners, Laf1 and Dgr2. Laf1 and Dgr2 are paralogues, both predicted to adopt a beta-propeller structure. The authors show that Laf1 and Dgr2 localize to ER-PM contact sites, and that this localization requires Lam2 and Lam4. Focusing on Lam2, the authors map the interaction domain with Laf1 to the two sterol-binding StART domains, and show that Lam2’s PH domain inhibits this interaction. Using a phosphomimetic mutant of Lam2, and an in vitro Ypl1 phosphorylation assay in vitro, the authors convincingly show that Ypk1-mediated phosphorylation of Lam2 inhibits its interaction with Laf1. This study is very well executed, and provides important and interesting new information on the regulation of two crucial lipid transfer proteins. I have only a few minor comments.
Detailed comments:
- Line 333-339. In Figure 1 Panel B, it is very striking that the intensity of the Lam2-mNG and Laf1-mNG puncta in the delta-tether strain is much more intense than in the WT control. Are protein levels the same in the two strains? What is the basis for suggesting these proteins self-associate in the delta-tether strain? At least for Lam2-mNG, there is a significant pool of the protein in round structures (vacuoles?), which is not seen in the delta-tether mutant. Have the authors investigated this aspect of the change in localization?
- Minor point: Could the authors define mNG.
- Lines 390-395, Figure 1 Panel D. Lower panel. What do the two bands correspond to? Are they phosphorylated and non-phosphorylated versions of Dgr2-mKate?
- In Figure 3 Panel A lower panel (line 469), which bands correspond to GST-Laf1 (pMT2) and GST-Dgr2 (pFR404) in the GST immunoblot? In Panel B (line 485), why are there multiple bands for Lam2(633-1259)-(His)6 (pFR408) and Lam2(849-1259)-488 (His)6?
- For some of the Western blots, I would like to see (as supplemental data) at least some of the original uncut blots. For example, Figure 2 has individual bands cut out. Many of the IB: His panels have too high a level of contrast (in particular Figure 3 Panel A, Figure 4 Panel D). These should be replaced.
- Lines 633-635. The authors state “… we suspect that the residues lining the central cavity in the Laf1 and Dgr2 ten-bladed -propellers selectively accommodate ergosterol.” This speculation seems to come out of nowhere. The authors have stated repeatedly that beta-propeller domains mediate protein-protein interactions. Why would they bind to ergosterol? The authors should point out that some beta-propeller domains been shown to directly bind phospholipids, although none have been shown to bind sterols, to my knowledge. Hence, to provide some justification for making the hypothesis that Laf1 and Dgr2 bind directly to sterols, it would be good to cite references to other lipid-binding beta-propeller domain proteins.
I believe that authors should provide some context when they present the results of Western blots. Especially in a case where a figure contains individual bands cut and pasted together (as in Figure 2 here), they should at least provide the original blot or blots from which the data came as supplementary data. I'm not sure this is necessary for all the blots, some have more information, but at least for Figure 2 I would say it is important.
Author Response
REVIEWER #1
Comments and Suggestions for Authors
—
The manuscript by M. Topolska et al. is an interesting and thorough study of the regulation of two sterol transport proteins important for maintaining the lipid composition of the PM. In eukaryotes, each cellular compartment has a unique lipid composition, maintained largely by lipid transfer at membrane contact sites. Two ER transmembrane proteins, Lam2 and Lam4, concentrate at ER-PM contact sites (CS) where they transfer sterol from its site of synthesis in the ER to the PM. The authors had previously identified Lam2 and Lam4 as direct targets of the Ypk1 protein kinase, a key regulator in the TORC2 pathway. Here they show that Ypk1 phosphorylation inhibits sterol transfer by Lam2 and Lam4, through regulation of Lam2/Lam4 interaction with two protein partners, Laf1 and Dgr2. Laf1 and Dgr2 are paralogues, both predicted to adopt a beta-propeller structure. The authors show that Laf1 and Dgr2 localize to ER-PM contact sites, and that this localization requires Lam2 and Lam4. Focusing on Lam2, the authors map the interaction domain with Laf1 to the two sterol-binding StART domains, and show that Lam2’s PH domain inhibits this interaction. Using a phosphomimetic mutant of Lam2, and an in vitro Ypk1 phosphorylation assay in vitro, the authors convincingly show that Ypk1-mediated phosphorylation of Lam2 inhibits its interaction with Laf1. This study is very well executed, and provides important and interesting new information on the regulation of two crucial lipid transfer proteins. I have only a few minor comments.
Action taken: We thank this referee for the very laudatory remarks on this work. We have addressed the specific comments, which these referee characterized as "minor," below.
Detailed comments:
- Line 333-339. In Figure 1, Panel B, it is very striking that the intensity of the Lam2-mNG and Laf1-mNG puncta in the delta-tether strain is much more intense than in the WT control. Are protein levels the same in the two strains? What is the basis for suggesting these proteins self-associate in the delta-tether strain? At least for Lam2-mNG, there is a significant pool of the protein in round structures (vacuoles?), which is not seen in the delta-tether mutant. Have the authors investigated this aspect of the change in localization?
Action taken: We have repeated the comparison of these two strains yet again, but using growth in YPD (rich) medium instead of minimal medium. When grown under this condition, there is no vacuole-associated Lam2-mNG. Careful inspection of the control cells shows numerous faint Lam2-mNG-containing dots (ER-PM contact sites) around the cell periphery, whereas the tether∆ hextuple mutant cells shows many fewer, but much brighter, Lam2-mNG-containing dots. That is the basis for suggesting that, in the absence of the six tether proteins, the Lam2-mNG dots remain at ER-PM junctions, but coalesce together, possibly via self-association [see also item #4 below].
- Minor point: Could the authors define mNG.
Action taken: The abbreviation mNG stands for the fluorescent protein mNeonGreen [Steiert F, Petrov EP, Schultz P, Schwille P, Weidemann T (2018) Photophysical behavior of mNeonGreen, an evolutionarily distant green fluorescent protein. Biophys. J. 114: 2419-2431]. It is first mentioned in the text (and the abbreviation defined) in the Materials and Methods section (at what is now line 281).
- Lines 390-395, Figure 1 Panel D. Lower panel. What do the two bands correspond to? Are they phosphorylated and non-phosphorylated versions of Dgr2-mKate?
Action taken: Yes. As stated already in the text, Dgr2 appears to be a phosphoprotein and a target of Ypk1, based on the multiple electrophoretic isoforms observed (Fig. 1D, lower panel) and on the fact that its sequence contains two Ypk1 consensus phospho-acceptor sites (now line 369ff).
- In Figure 3 Panel A lower panel (line 469), which bands correspond to GST-Laf1 (pMT2) and GST-Dgr2 (pFR404) in the GST immunoblot? In Panel B (line 485), why are there multiple bands for Lam2(633-1259)-(His)6 (pFR408) and Lam2(849-1259)-488(His)6?
Action taken: In Fig. 3A, the uppermost (slowest mobility) band in the anti-GST immunoblot is the full-length GST-Laf1 or GST-Dgr2 constructs, respectively, in the lanes so indicated; and, the figure has now been modified to indicate the corresponding bands. The faster mobility species are degradation products of the full-length GST-Laf1 and GST-Dgr2 constructs. Similarly, in Fig. 3B, the major bands in the anti-His immunoblot of the input are Lam2(482-1259)-(His)6, Lam2(633-1259)-(His)6 and Lam2(633-1259)-(His)6. For Lam2(482-1259)-(His)6 and Lam2(633-1259)-(His)6, the faster mobility species are degradation products of the full-length constructs, whereas the upper band in the Lam2(633-1259)-(His)6 is due to a minor amount of spill over (i.e. contamination) from the neighboring Lam2(482-1259)-(His)6-containing lane [and, by the way, its robust appearance in the pull-down provides some evidence for Lam2-Lam2 self-association (see item #1 above)].
- For some of the Western blots, I would like to see (as supplemental data) at least some of the original uncut blots. For example, Figure 2 has individual bands cut out. Many of the IB: His panels have too high a level of contrast (in particular Figure 3 Panel A, Figure 4 Panel D). These should be replaced.
Action taken: At the time of manuscript submission, it was requested that the raw / original data for each of the immunoblots be provided as a separate entry file and we did so. I don't know why the referee did not have access to that information during the review process. Showing the relevant bands in the main figures, rather than entire gels, is standard practice; and, none of the exposures shown are at a "too high a level of contrast." Therefore, we have not altered any of the original data shown (and, by the way, Reviewer #2 raised no objections whatsoever with regard to any immunoblot).
- Lines 633-635. The authors state “… we suspect that the residues lining the central cavity in the Laf1 and Dgr2 ten-bladed b-propellers selectively accommodate ergosterol.” This speculation seems to come out of nowhere. The authors have stated repeatedly that beta-propeller domains mediate protein-protein interactions. Why would they bind to ergosterol? The authors should point out that some beta-propeller domains been shown to directly bind phospholipids, although none have been shown to bind sterols, to my knowledge. Hence, to provide some justification for making the hypothesis that Laf1 and Dgr2 bind directly to sterols, it would be good to cite references to other lipid-binding beta-propeller domain proteins.
Action taken: We are a little taken aback by this comment because it is clearly explained in the Discussion that the b-propellers in Laf1 and Dgr2 are unusual in having ten predicted WD-40 motifs (instead of the more typical seven), that at least one other ten-bladed b-propeller protein is able to bind a small molecule (a peptide hormone) in its central cavity, and, most importantly, we cited a very recent published paper reporting that Laf1 is indeed a sterol-binding protein [Ref. 68]. So, we feel that this specific concern of the referee has already been adequately addressed.
- I believe that authors should provide some context when they present the results of Western blots. Especially in a case where a figure contains individual bands cut and pasted together (as in Figure 2 here), they should at least provide the original blot or blots from which the data came as supplementary data. I'm not sure this is necessary for all the blots, some have more information, but at least for Figure 2 I would say it is important.
Action taken: Once again, at the time of manuscript submission, it was requested that the raw / original images for each of the immunoblots be provided as a separate file entry and we did so. I don't know why the referee did not have access to that information during the review process. Showing the relevant bands in the main figures (rather than entire gels) is standard practice and none of the exposures shown are at a "too high a level of contrast." Therefore, we have not altered any of the original data shown (moreover, Reviewer #2 raised no objections whatsoever with regard to any of the immunoblots).
Reviewer 2 Report
In this manuscript the authors evaluated the potential role of Ypk1-mediated phosphorylation in regulation of the sterol transport by Lam2 and Lam4 at ER-PM CSs. Previous work of these authors have shown that Ypk1-mediated phosphorylation inhibited the ability of Lam2 and Lam4 to promote retrograde transport of sterol from the PM to the ER, thereby retaining ergosterol in the PM under conditions that stimulate TORC2-mediated activation of Ypk1. That regulation has be suggested to enable the cell survival under stress conditions. In the present work using several Saccharomyces cerevisiae strains the authors examined whether Ypk1-specific phosphorylation can affect the localization or stability of Lam2 and Lam4, and how phosphorylation of Lam2 by Ypk1 may influence the ability of that protein to interact with Laf1. The experiments were conducted in vivo and in vitro using the recombinant proteins. The presentation of the data is clear and the author`s conclusion are valid. In my opinion the work should be accepted with only minor revisions:
- Subheadings should be used for divisions of the titled sections of the manuscript (lines 305, 456, 519)
- For a better presentation, colocalization of the examined proteins could be shown by the arrows in all panels of Fig.1 and in Fig. 4, panels B and C.
Author Response
REVIEWER #2
Comments and Suggestions for Authors:
In this manuscript the authors evaluated the potential role of Ypk1-mediated phosphorylation in regulation of the sterol transport by Lam2 and Lam4 at ER-PM CSs. Previous work of these authors have shown that Ypk1-mediated phosphorylation inhibited the ability of Lam2 and Lam4 to promote retrograde transport of sterol from the PM to the ER, thereby retaining ergosterol in the PM under conditions that stimulate TORC2-mediated activation of Ypk1. That regulation has be suggested to enable the cell survival under stress conditions. In the present work using several Saccharomyces cerevisiae strains the authors examined whether Ypk1-specific phosphorylation can affect the localization or stability of Lam2 and Lam4, and how phosphorylation of Lam2 by Ypk1 may influence the ability of that protein to interact with Laf1. The experiments were conducted in vivo and in vitro using the recombinant proteins. The presentation of the data is clear and the author`s conclusion are valid. In my opinion the work should be accepted with only minor revisions:
Action taken: We thank this referee for the very kind remarks about this work. We have addressed the two minor comments of this referee below.
- Subheadings should be used for divisions of the titled sections of the manuscript (lines 305, 456, 519)
Action taken: We think that the sub-section headings are already appropriate and sufficient, as is.
- For a better presentation, colocalization of the examined proteins could be shown by the arrows in all panels of Fig.1 and in Fig. 4, panels B and C.
Action taken: As requested by this referee, arrows have been added to Fig. 1. showing, in the merge, representative dots where GFP-Lam2 and Laf1-mKate co-localize. We feel, however, that it was not beneficial to add any arrows to Figs. 4B and C because it is already obvious that GFP-Lam2 and Laf1-mKate are still co-localizing to the same dots in these figures.